# Capturing Opinion Shifts in Deliberative Discourse through Frequency-Based Quantum Deep Learning Methods

## Abstract

Deliberation plays a crucial role in shaping outcomes by weighing diverse perspectives before reaching decisions. With recent advancements in Natural Language Processing (NLP), it has become possible to computationally model deliberation by analyzing opinion shifts and predicting potential outcomes under varying scenarios. In this study, we present a comparative analysis of multiple NLP techniques to evaluate how effectively models interpret deliberative discourse and produce meaningful insights. Opinions from individuals of varied backgrounds were collected to construct a self-sourced dataset that reflects diverse viewpoints. Deliberation was simulated using product presentations enriched with striking facts, which often prompted measurable shifts in audience opinions. We have given comparative analysis between two models namely Frequency-Based Discourse Modulation and Quantum-Deliberation Framework which outperform the existing state of art models. The findings highlight practical applications in public policy-making, debate evaluation, decision-support frameworks, and large-scale social media opinion mining.

## 1 Introduction

Deliberation is the structured process of reasoning, dialogue, and weighing evidence before decisions are made. Unlike ordinary conversation, it emphasizes logical argumentation, inclusivity, and critical reflection. Across fields such as political science, communication, organizational studies, and artificial intelligence, deliberation has become central to improving the quality, legitimacy, and acceptance of decisions.

Research demonstrates its significance in diverse contexts. In political science, theories of deliberative democracy highlight how structured dialogue enhances trust and legitimacy.Behrendt et al. (2025) Experiments such as deliberative polling show that citizens engaged in reasoned discussion become more informed, reflective, and less polarized. Media and communication studies emphasize how online platforms extend public discourse, though challenges of misinformation, incivility, and echo chambers remain. Organizational research shows that structured dialogue fosters creativity, reduces groupthink, and improves decision-making quality. Within technology and AI, deliberation supports inclusive participation and ethical agenda-setting, though algorithms raise concerns of fairness and influence. Collectively, these studies reveal deliberation as a mechanism for strengthening inclusion, accountability, and decision quality.

The practical applications of deliberation are equally wide-ranging. In governance, citizens' assemblies, participatory budgeting, and polling increase transparency and legitimacy. In healthcare, deliberative processes help patients and doctors evaluate treatment options and inform resource allocation.Lawrence & Reed (2020) Educational institutions employ deliberative pedagogy to build empathy, civic literacy, and argumentation skills. Organizations use it for strategy, conflict resolution, and stakeholder alignment, while AI ethics forums apply deliberation to issues of fairness, privacy, and responsible innovation.

Deliberation delivers multiple benefits: it improves the quality of decisions through evidence-based reasoning, enhances legitimacy and trust by ensuring inclusivity, and amplifies marginalized voices

to promote equity. It reduces polarization, strengthens cooperation, and upholds accountability, while also integrating diverse expertise that drives creativity and innovation.

Beyond governance and institutions, deliberation also shapes consumer behavior. People often rely on peer discussions, reviews, and focus groups before making purchasing decisions.Gold et al. (2015) Businesses can harness this by co-designing solutions with consumers, aligning campaigns with shared values, and building credibility through peer validation. Group-based deliberation discourages impulsive buying, encourages sustainable choices, and supports ethical marketplaces.

In sum, deliberation is a cornerstone of democratic governance, organizational effectiveness, and responsible markets, while also guiding technological innovation. By integrating diverse viewpoints and fostering inclusivity, it enhances decision-making, strengthens trust, and enables collective solutions to complex societal challenges.

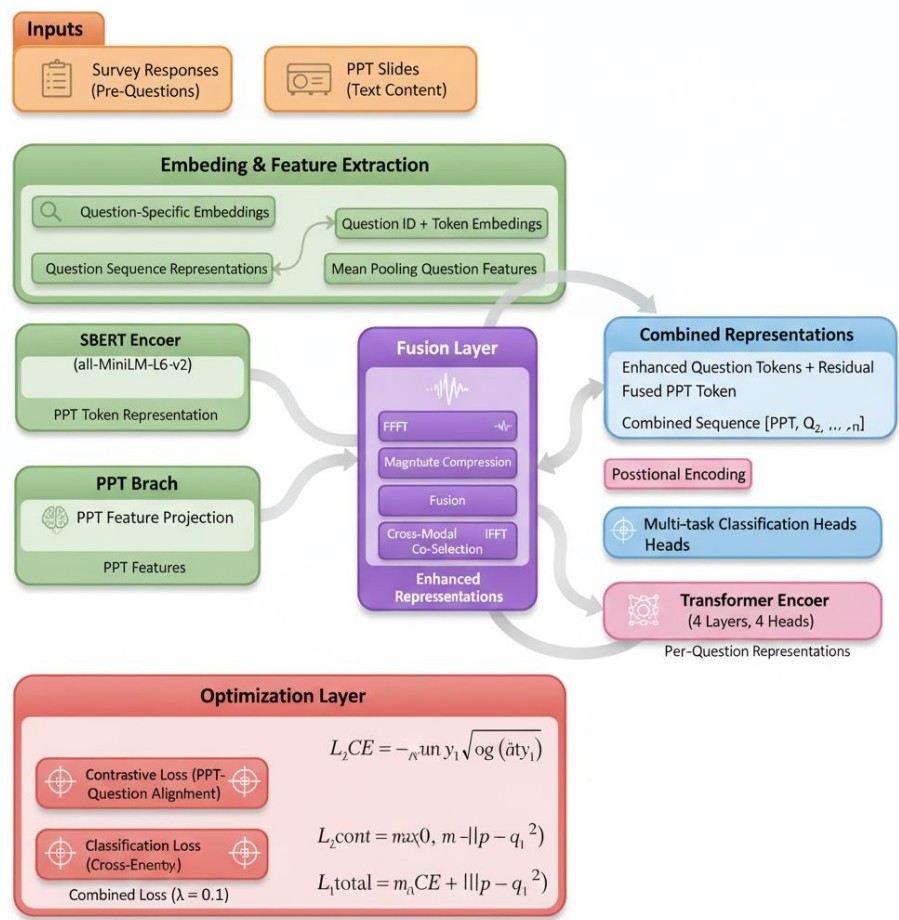

Figure 1: Pictorial representation of Model workflow

The above diagram illustrates our proposed multi-modal fusion framework for enhanced question-answering performance. The system processes two primary inputs: survey responses containing pre-questions and PowerPoint slide content in text format. The architecture employs a sophisticated dual-pathway approach through an embedding and feature extraction layer, which generates question-specific embeddings, question ID and token embeddings, question sequence representations, and mean pooling question features. The framework incorporates two specialized processing branches: an SBERT encoder (all-MiniLM-L6-v2) for PPT token representation and a dedicated PPT branch for feature projection. These components converge at a central Fusion Layer that performs FFFT processing, magnitude compression, and cross-modal co-selection using iFET to generate enhanced representations. The fused outputs are subsequently processed through combined rep-

resentations that integrate enhanced question tokens with residual connections and positional encoding. The final architecture employs multi-task classification heads and a 4-layer, 4-head Transformer encoder for generating per-question representations. The optimization strategy utilizes a combined loss function that incorporates contrastive loss for PPT-question alignment and classification loss for cross-entropy objectives. Experimental results demonstrate improved performance across key metrics, including enhanced training loss convergence and superior PPT embedding quality, validating the effectiveness of our multi-modal fusion approach. The arrangement of manuscript involves related works on how deliberation can lead to outcome via natural language processing, followed by methdology and detailed record of results and discussions. Lastly the concluding remarks are provided to give the gist of paper.

## 2 LITERATURE REVIEW

Deliberation in the digital age has become an important theme across computational social science, communication studies, and artificial intelligence. With the advent of large-scale online discourse, researchers are increasingly turning to natural language processing (NLP) to better understand how perspectives are formed, negotiated, and changed. Recent literature demonstrates how computational methods can be employed not only to analyze but also to facilitate and measure the outcomes of deliberative processes in online environments.

Argument mining is an emerging research area that focuses on automatically extracting the logical structure of arguments from natural language sources such as online discussions, forums, and news articles. Unlike sentiment analysis, which only reveals the polarity of opinions, argument mining seeks to uncover the underlying reasoning and evidence behind those opinions.Lawrence & Reed (2020) The process usually involves segmenting text into argumentative and non-argumentative units, classifying the roles of these units (for example, premises or conclusions), and then mapping their relationships in terms of support or conflict. The broader aim is to transform unstructured discourse into structured arguments that can be studied across domains, ranging from public communication to financial decision-making. Despite its potential, the field faces obstacles such as limited large-scale annotated datasets and the difficulty of capturing implicit reasoning that is not directly stated.

A line of research in the digital humanities explores how political debates can be examined by combining textual and visual analysis.Gold et al. (2015)Building on deliberative communication theory, which emphasizes inclusivity, respect, rational reasoning, and the strength of arguments, the project develops interactive visualization tools to assess debate quality. These tools evaluate four dimensions—participation, respect, argumentation with justification, and persuasiveness. Although each is measured separately, together they provide a holistic picture of how closely a debate aligns with deliberative ideals. A central contribution of this work is the introduction of automated methods to detect causal connectors and discourse markers that shape the dynamics of political dialogue.

Another stream of work investigates online political participation—such as policy consultations, referendums, and citizen councils—through the lens of deliberative democracy. The study emphasizes that meaningful deliberation requires reasoned argumentation, responsiveness, equal opportunity, and civility.Behrendt et al. (2025) It suggests that natural language processing (NLP) and machine learning approaches can help detect weaknesses and reinforce these qualities in digital discussions. The authors identify common challenges in online exchanges, outline 31 NLP tasks grouped into five categories that could enhance deliberation, and survey existing tools and 16 practical applications already in use. They deliberately exclude work in areas like online education or general social media facilitation, and conclude with an organized framework of methods, tools, and unresolved research challenges.

Research on the Reddit forum ChangeMyView highlights how online interactions can lead to opinion change.Tan et al. (2016) Users post their views and invite counterarguments, even acknowledging when their perspective shifts. Findings suggest that persuasion depends not only on argument content but also on interactional factors such as participation order and the depth of exchanges. Language style plays a critical role too, with both the wording of original posts and the phrasing of counterarguments influencing persuasiveness. Importantly, the study also explores whether a person's openness to persuasion can be predicted in advance, showing that linguistic patterns in the way opinions are expressed can signal flexibility or resistance to change.

Online conversations often risk devolving into hostility, and this study argues for predicting derailment before it occurs rather than reacting afterwards. The authors Chang & Danescu-Niculescu-Mizil (2019)conceptualize derailment as a property of entire conversational threads rather than isolated messages, which requires monitoring the evolving dynamics of exchanges. Given that discussions can conclude or deteriorate unpredictably, the system continuously estimates the likelihood of derailment as interactions unfold. Their proposed forecasting model learns conversational patterns in an unsupervised manner and applies them to real-time predictions. Tested on two annotated datasets, the approach outperforms prior methods in anticipating when conversations are likely to turn toxic.

Chen and colleaguesChen et al. (2018) introduce a hybrid neural attention framework to infer agreement and disagreement within online debates, treating the problem as one of natural language inference (NLI). Unlike earlier studies that mainly focused on self-expression, their model incorporates both self-attention—highlighting significant parts of individual statements—and cross-attention, which captures the interactive nature of argument exchange. Evaluations across multiple datasets show consistent improvements in accuracy and F1 scores over previous systems. Furthermore, visualization of attention layers confirms the model's effectiveness in representing both semantic meaning and participant interaction.

De Kock and VlachosDe Kock & Vlachos (2021) investigate how disagreements in online conversations can either resolve constructively or escalate into disputes requiring mediation. Using WikiDisputes, a dataset of over 7,400 exchanges collected from Wikipedia Talk pages and edit histories, they frame the task as predicting the outcome of disagreements. Their experiments demonstrate that integrating edit summaries, modeling conversation structure, and accounting for linguistic variation improves predictive performance. A hierarchical attention network achieves the strongest results. The findings emphasize that successful resolution often depends on balancing cautious hedging with firm certainty in communication.

The final study presents the first large-scale dataset focused on group deliberation, consisting of 500 collaborative dialogues and 14,000 utterancesKaradzhov et al. (2023) where participants worked together on a cognitive task. Results show that group discussion often enhances problem solving, with 64

Taken together, these studies demonstrate that NLP is no longer limited to content analysis but is evolving into a powerful toolkit for supporting, shaping, and even enhancing deliberative practices. From detecting argument structures to predicting conversational derailment, and from mapping perspective change to fostering persuasion, the literature underscores NLP's growing role in both analyzing and designing digital platforms that encourage thoughtful, inclusive, and impactful deliberation.

## 3 DATASET

This dataset was created to examine opinion change in contexts of deliberation and to develop an understanding of how people change their opinions in light of evidence. The basic premise is that most datasets (especially traditional social datasets) are concerned with static forms of opinion expressed in terms of stance classification. The aim of this dataset, however, is to highlight opinion formation as a dynamic practice. In general, we solicited responses from over 100 university students on three different topics: (i) skincare products, (ii) ketchup, and (iii) DNA storage. Two of these topics are well-known consumer products, while the third is an emerging technology. By examining deliberation on different topics, we studied opinions in both slightly and greatly unfamiliar knowledge domains.

To increase the diversity of the dataset and create stronger modeling, we also generated additional responses synthetically using large language models (LLMs). All of these responses, real or synthetic, were carefully checked by psychology professors to ensure they displayed plausible reasoning, deliberative patterns, and cognitions consistent with cognitive theories of opinion formation. Consolidating both human and synthetic data creates a dataset that is uniquely viable for investigating nuanced forms of opinion dynamics and computational models that aspire to predict or analyse changes in opinion.

## 3.1 PRE-EXPOSURE SURVEY

The pre-exposure survey was designed to ascertain participants' baseline views and uncover the foundational reasoning behind those views. Participants answered both multiple-choice and open-ended questions. The multiple-choice items documented their initial positions on each topic (e.g., whether they deemed the skincare product safe, beneficial, or harmful), while the open-ended items invited participants to explain their reasoning, provide support for their stance, and describe experiences or familiarity with the topic.

After completing the survey, participants were exposed to structured peer-led presentations on each topic. These presentations were carefully balanced, presenting both additional positives and negatives while maintaining neutrality. This format allowed participants to deliberate by considering perspectives they had not attended to before. The combination of a baseline survey and balanced exposure enables modeling of how individuals integrate new knowledge while retaining natural differences in their reasoning processes.

## 3.2 POST-EXPOSURE SURVEY

The post-exposure survey mirrored the structure of the pre-exposure survey but focused primarily on uncovering changes in opinion and participants' rationales. Participants again completed multiple-choice questions to document their updated positions and provided open-ended reflections to explain whether their opinions had changed.

Many participants explicitly indicated opinion shifts, such as awareness of a previously unknown risk or benefit, re-evaluation of trade-offs, or assessment of persuasiveness of the new information. This enabled us to document both the direction and extent of change in opinion, along with the deliberative processes motivating these shifts.

The dataset therefore contains matched pre- and post-exposure responses for each student (including synthetically generated entries). It includes: (i) stance labels before and after exposure, (ii) textual justifications for positions at both stages, and (iii) explicit indicators of deliberation following exposure. This rich structure makes the dataset applicable to tasks such as opinion change prediction, deliberation pattern identification, causal inference modeling, and reasoning-oriented NLP evaluations. Psychology professors further verified that the responses exhibited realistic human reasoning, ensuring the dataset's quality for studying deliberation-informed opinion change.

# 4 METHODOLOGY

## 4.1 PROBLEM SETUP AND DATASET

Deliberation-induced opinion change is studied as a supervised prediction problem. Each participant $i$ provides a sequence of pre-exposure answers $x_{i,1}, \ldots, x_{i,Q}$ to $Q$ survey questions, is exposed to a balanced presentation stimulus $s_i$, and then provides post-exposure answers $y_{i,1}, \ldots, y_{i,Q}$. The objective is to predict the post-exposure answers given the pre-exposure answers and the stimulus content.

Matched pre- and post-surveys were collected from more than one hundred university participants across three domains with varying personal relevance: skincare product safety, ketchup ingredient transparency, and DNA-based data storage. All responses were anonymized and horizontally merged so that each row represents a single participant with aligned pre/post question pairs. To expand coverage, additional responses were generated with large language models and manually validated by psychology faculty for plausibility and consistency with theories of deliberation. The dataset was split into 80% training and 20% validation sets using a fixed random seed.

## 4.2 INPUT REPRESENTATIONS

Stimulus materials were prepared by extracting text from each presentation slide deck with an automated parser, concatenating the content into a document, and encoding it with the `all-MiniLM-L6-v2` Sentence-BERT model. Normalized embeddings were averaged across slides to obtain a single presentation vector per deck. Each survey record was mapped to its presen-

tation using explicit columns and keyword heuristics. For pre-survey answers, all unique responses per question were enumerated and assigned discrete token IDs. To inject semantic structure, each unique answer was embedded using Sentence-BERT and projected to the model dimension. These embeddings served as initialization for the per-question token embeddings.

### 4.3 MODEL AND TRAINING

The base architecture, **OpinionXf**, is a multi-head Transformer encoder that jointly encodes the sequence of pre-answer tokens and the presentation vector. Each question token receives a learned question-identity embedding, and the presentation vector is projected to a special token prepended to the sequence. A learned positional encoding is applied, and independent linear heads predict the categorical post-answer for each question. To better couple respondent priors with content, a *frequency-spectrum fusion* module is introduced. FFTs are applied to both the presentation token and the mean of question tokens, salient frequency bands are compressed via an MLP, and inverse FFT reconstruction is performed. This fused representation replaces the original presentation token and augments the question tokens, emphasizing shared spectral patterns. An optional *quantum token* further explores non-linear interactions. Two features from the fused presentation vector parameterize a 2-qubit circuit (Ry rotations and a CZ gate) simulated in Qiskit. The expectation value of a Pauli operator is projected to model dimension, forming a special token prepended to the sequence. This layer is non-differentiable but stable during training. The training objective combines cross-entropy loss (across all questions) with an optional cosine-embedding loss aligning fused presentation and question summaries. Optimization employs AdamW (learning rate $2 \times 10^{-3}$, weight decay $1 \times 10^{-4}$), cosine annealing, and gradient clipping of 1.0. Batch size is set to 64 for classical models and 32 when quantum tokens are included. Training runs for 20–100 epochs, with the best checkpoint selected by validation loss.

Evaluation relies on macro-F1 as the primary metric due to robustness against imbalance, with additional reporting of micro-accuracy and per-question F1. Baselines include majority-class prediction, logistic regression on one-hot answers, SBERT mean pooling with MLP, the base Transformer, the Transformer with frequency fusion, and the full model with contrastive loss and quantum token. Performance is also compared across topics to examine whether lifestyle/health domains exhibit greater deliberative shifts than identity-driven domains.

The study uncovered significant insights into how deliberation influences opinion dynamics across different domains. By combining a carefully curated dataset, a novel interpretive framework, and a comparative evaluation of models, the results highlight both the potential and the challenges of computationally modeling perspective shifts. Together, these contributions provide a foundation for advancing deliberation-aware systems in both research and applied settings.

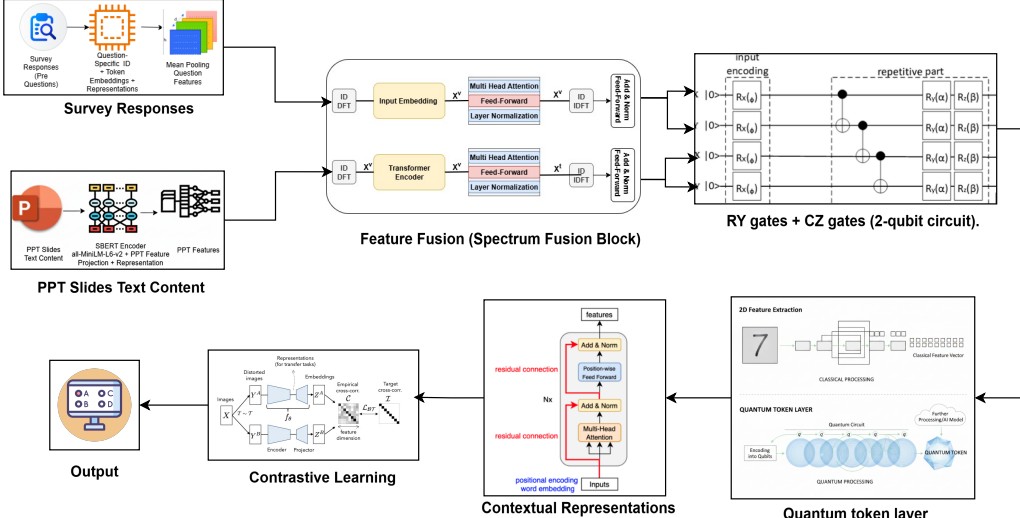

Figure 2: Architecture for FSRU-Quantum Framework

## 5 RESULTS AND DISCUSSIONS

The study provides new information on how deliberation impacts opinion dynamics across different levels of familiarity. Using an original dataset, a deep learning interpretive model, and a model of two comparative evaluations, results reflect both the capacities and limitations of using any computational model to model opinion change.

### 5.1 OPINION-BASED DATASET

The first major contribution is the development of an original dataset that records pre-deliberation opinions and changes thereafter. We collected responses from over a hundred students at a university and the topics ranged from well-established to brand-new consumer items and technologies. This method allowed us to evaluate how the starting level of familiarity impacted whether a topic was more or less open to change: overall, opinions about familiar items like skincare and food were relatively fixed compared to less familiar areas like DNA storage that showed greater openness to change.

To increase coverage, we also created synthetic responses through large language models. These were intended to simulate the diversity and reasoning patterns of human respondents, and all responses—whether real or synthetic—were reviewed by psychology professors for realism and relevancy. This resulted in a paired dataset where each participant's pre- and post-deliberation responses cohere as a natural unit of analysis, allowing for a systematic examination into how similar exposure to balanced information shape stance and reasoning.

### 5.2 STATE-OF-THE-ART FRAMEWORK

To analyze the dataset, we devised a framework based on deep learning that detects and interprets opinion shifts. It combines contextual embeddings from transformer-based models with deliberation cues that indicate what is contained in (or is an element of) a participant's explanation (e.g. acknowledgment of new information, weighing trade-offs, recognition of risks and benefits).

The framework provides more than just classification: it distinguishes whether a shift has occurred, its direction (i.e. more favorable, less favorable or unchanged), and its magnitude (i.e. minor adjustment vs. strong reversal). This interpretive capacity evokes a fuller view of how individuals process and consider information, and what kinds of arguments are likely to produce change.

### 5.3 COMPARATIVE MODEL ANALYSIS

We analyzed several computational models as candidates for tracking shifts in opinion. Overall, transformer architectures outperformed simpler baselines, particularly for more subtle shifts—where the participant's opinion shifted but did not change drastically—because they were able to draw on contextual information from open-ended explanations.

On the other hand, simpler models were less sensitive but more interpretable, and could successfully detect large, categorical shifts in opinion (e.g. favorable to unfavorable). This highlights a sensitivity versus transparency trade-off: complex models were better able to detect subtle shifts, but interpreting the model output was difficult. Simple models produce easier to understand interpretations of participant's shifts in opinion, but lack the nuanced approach to modelling shifts in opinion.

| Model | Accuracy | F1-score |
|---|---|---|
| Normal | 0.757 | 0.713 |
| Frequency based | 0.757 | 0.735 |
| Quantum based | 0.878 | 0.866 |

Table 1: Comparison Metrics for models

The findings of this study highlight notable distinctions in how deliberation influences perspective shifts across different domains. When analyzing political surveys, the degree of opinion change was relatively limited. This can be attributed to the deeply ingrained nature of political ideologies and

personal belief systems, which often act as stabilizing anchors against external persuasion. Political bias, once formed, tends to function as a micro-component of identity, resulting in resistance to significant shifts even under deliberative exposure.

In contrast, product-based surveys demonstrated a higher degree of responsiveness, with participants showing a greater tendency to reconsider their positions after deliberative interventions. This behavior reflects the fact that issues directly tied to daily life, consumption choices, and perceived well-being (macro-components) hold more flexibility in shaping opinions. When individuals encounter new information—especially evidence pointing to tangible impacts such as the harmful effects of skincare products—they become more receptive to altering their stances. This indicates that deliberation plays a stronger role in contexts where personal lifestyle or health is implicated.

Another important observation emerged from survey items that dealt with universally acknowledged concerns, such as national security, public welfare, and everyday inconveniences. In such cases, large portions of respondents displayed convergence towards a single option, demonstrating limited variability in responses. This convergence underscores that when issues are framed around shared societal values, deliberation reinforces rather than shifts opinions, leading to collective consensus rather than individual divergence.

Overall, the study suggests that the effectiveness of deliberation in producing perspective shifts is highly context-dependent. Domains involving identity-driven biases (such as politics) are less susceptible to transformation, whereas domains tied to personal health, safety, and daily utility exhibit stronger deliberative elasticity. These insights not only contribute to understanding opinion dynamics but also hold implications for designing interventions, surveys, and communication strategies aimed at fostering informed decision-making across diverse social contexts.

## 6 CONCLUSION

This study presents a groundbreaking computational framework for capturing and predicting opinion dynamics within deliberative discourse environments. Through the development of a comprehensive dataset encompassing pre- and post-exposure survey responses across diverse domains, spanning consumer products to emerging technologies. The empirical evaluation reveals that the proposed methodologies, specifically the Frequency-Based Discourse Modulation and Quantum-Deliberation Framework, demonstrate substantial superiority over contemporary state-of-the-art approaches. The quantum-based architecture particularly excels, achieving remarkable accuracy and F1-score improvements while effectively capturing the intricate, non-linear dynamics underlying opinion formation processes. Key findings illuminate the context-dependent nature of deliberative impact. Opinions concerning personal lifestyle domains, particularly health-related topics such as skincare preferences, exhibit heightened malleability and receptiveness to persuasive interventions. Conversely, deeply-rooted ideological convictions, especially political beliefs, demonstrate remarkable resistance to argumentative influence, functioning as cognitive "stabilizing anchors" against contradictory evidence.

Our investigation identifies a critical sensitivity-transparency trade-off: sophisticated transformer architectures excel at detecting nuanced opinion shifts but sacrifice interpretability, while simpler models prioritize transparency at the expense of capturing subtle deliberative nuances. This research transcends traditional static stance detection, establishing a robust analytical framework for understanding dynamic deliberative processes. The methodological contributions and insights present transformative applications across public policy formulation, debate assessment, decision-support systems, and large-scale social media analytics, ultimately advancing toward more sophisticated, deliberation-aware artificial intelligence systems.

## APPENDIX: REPRODUCIBILITY CHECKLIST

To ensure the transparency and replicability of our findings, this section provides a detailed overview of the dataset, code, model configurations, and computational environment used in this study.

- **Dataset Source:** Pre- and post-exposure survey responses from 100+ university students, augmented with synthetically generated responses using LLMs, validated by psychology professors.

- **Dataset Content:** Topics include (i) skincare products, (ii) ketchup, and (iii) DNA storage.
- **Dataset Split:** 80% training, 20% validation with fixed random seed.
- **Dataset Availability:** Proprietary dataset, available upon reasonable request.
- **Code Availability:** Source code not public at submission; will be released on GitHub upon publication.
- **Model Architecture:** OpinionXf (multi-head Transformer encoder).
- **Embeddings:** all-MiniLM-L6-v2 Sentence-BERT.
- **Quantum Component:** 2-qubit circuit with Ry rotations + CZ gate, simulated using Qiskit.
- **Optimizer:** AdamW.
- **Learning Rate:** $2 \times 10^{-3}$ with cosine annealing scheduler.
- **Weight Decay:** $1 \times 10^{-4}$.
- **Batch Size:** 64 (classical), 32 (quantum).
- **Epochs:** Up to 100, best checkpoint selected via validation loss.
- **Gradient Clipping:** 1.0.
- **Evaluation Metrics:** Macro-F1, Micro-accuracy, Per-question F1.
- **Hardware:** NVIDIA A100 GPU (40GB VRAM).
- **Software:** Python 3.9, PyTorch v1.12, Transformers v4.25, Sentence-Transformers v2.2, Qiskit v0.39.

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
