# OpenReview forum: "Capturing Opinion Shifts in Deliberative Discourse through Frequency-based Quantum deep learning methods"
_ICLR.cc/2026/Conference — Submitted to ICLR 2026_

### Official Review · Reviewer_iS4s · 2025-10-30

**Soundness:** 1
**Presentation:** 1
**Contribution:** 1
**Rating:** 2
**Confidence:** 4

**Summary:**

The paper states to present a comparative analysis of multiple NLP techniques, evaluating how effectively models interpret deliberative discourse and produce meaningful insights. The authors collected a dataset of opinions on three different topics, surveying participants before and after they were confronted with information about the topics in the form of PowerPoint presentations. These surveys capture the participants' opinion shifts. They then trained two different Transformer-based models to predict these shifts based on the answers given in the pre-survey and the PowerPoint text.

**Strengths:**

The topic of deliberation and the use of NLP to analyze and support dialogue is highly relevant.

**Weaknesses:**

The goal of the paper is unclear. The related work discussed does not seem to be related to the paper's overall topic. The dataset is supposed to capture shifts in opinion based on deliberation between participants, yet there is no interaction between participants; only confrontation with the source material.
Many important references are missing. For example:

- SBERT: Nils Reimers and Iryna Gurevych. 2019. Sentence-BERT: Sentence Embeddings using Siamese BERT-Networks. In Proceedings of the 2019 Conference on Empirical Methods in Natural Language Processing and the 9th International Joint Conference on Natural Language Processing (EMNLP-IJCNLP), pages 3982–3992, Hong Kong, China. Association for Computational Linguistics.
- Transformer: Vaswani, A., Shazeer, N., Parmar, N., Uszkoreit, J., Jones, L., Gomez, A. N., ... & Polosukhin, I. (2017). Attention is all you need. Advances in neural information processing systems, 30.
- Studies mentioned in the introduction

The model description is inadequate. It lacks formulas and only provides a description of layers and operations.
The structure of the paper is unclear. The details of the experiment are unclear. How many participants took part? How were the topics chosen? Why were additional answers generated by LLMs? How do the models predict opinion shifts? Only based on the stance of participants?

**Questions:**

see above

---

### Official Review · Reviewer_919g · 2025-10-31

**Soundness:** 2
**Presentation:** 2
**Contribution:** 2
**Rating:** 2
**Confidence:** 1

**Summary:**

This paper presents a computational framework for modelling and predicting opinion shifts within deliberative discourse. The authors introduce a new, self-sourced dataset collected from university students, supplemented with synthetically generated data, to capture opinions before and after exposure to a balanced presentation on various topics. The core of the contribution is a comparative analysis of two proposed models: a "Frequency-Based Discourse Modulation" model and a "Quantum-Deliberation Framework," both built upon a Transformer-based architecture named OpinionXf. The authors claim that these models, particularly the quantum-inspired variant, significantly outperform baseline approaches in predicting post-deliberation opinion stances, thereby offering a more nuanced way to analyse the dynamics of persuasion and belief change.

**Strengths:**

1. Relevant and Important Problem: The paper addresses the challenging and highly relevant problem of computationally modelling opinion dynamics. Understanding how deliberation leads to opinion shifts has significant implications for computational social science, political science, and AI ethics. The framing of the problem is a clear strength.
2. Novel Dataset Collection: The effort to create a new dataset specifically for this task is commendable. The pre- and post-exposure structure, capturing matched pairs of responses, is well-suited for studying opinion change. This is a valuable contribution, as such datasets are scarce.
3. Novelty of Approach: The paper proposes conceptually interesting ideas by integrating frequency-domain analysis (via FFT) and quantum-inspired computing into a standard Transformer architecture. This multi-modal fusion approach, aiming to capture complex interactions between a person's prior beliefs and new information, is original.

**Weaknesses:**

Despite the intriguing premise, the paper suffers from several significant weaknesses that undermine its central claims and contribution.
1. Lack of Clarity and Methodological Detail: The paper's narrative is often confusing, and the description of the proposed methods lacks the necessary detail for reproducibility and critical evaluation.
The writing style is convoluted, with several typos and awkward phrases that hinder comprehension. The methodological section is particularly opaque. The description of the "Quantum-Deliberation Framework" is too high-level. It states that a 2-qubit circuit is parameterised and simulated, but the process is not explained with sufficient mathematical rigour. Crucially, the paper mentions that "This layer is non-differentiable but stable during training," without explaining how gradients are handled or how this component is effectively integrated into an end-to-end deep learning model. This is a critical omission that makes the reported results difficult to trust.
2. Insufficient Experimental Scale and Validation: The empirical foundation of the study is weak.
The dataset, based on "over 100 university students," is very small. Although augmented with synthetic data, the paper lacks sufficient detail on the generation process, the potential biases introduced, or how the model's performance differs between real and synthetic samples. The generalizability of findings from such a limited and potentially homogeneous sample (university students) is a significant concern.
The massive performance leap reported for the "Quantum-based" model in Table 1 (F1-score from 0.735 to 0.866) is an extraordinary claim. Given the small dataset and the vague methodological description, this result requires much more substantial evidence and ablation studies to be convincing.
3. Inadequate Comparison to Baselines: The paper fails to properly contextualise its performance by comparing against baselines.
The evaluation is missing a crucial and powerful baseline from the current large generative models. The task of predicting post-exposure answers could be framed as a conditional generation or a multiple-choice question-answering problem for models like GPT-5 or Qwen in a zero-shot or few-shot setting.

**Questions:**

1. Could you please provide a detailed mathematical formulation for the quantum token generation and a detailed explanation of the format and examples for input, output, and intermediate results of every component?
2. Regarding the dataset: Could you elaborate on the synthetic data generation process? What LLM was used, what were the prompts, and what steps were taken to ensure that the synthetic data does not simply overfit to specific patterns that your proposed model is designed to capture? Have you conducted any analysis on the statistical significance of your results, given the small size of the human-generated portion of the dataset?
3. Could you clarify the distinction between the "Normal" model in Table 1 and the "base architecture, OpinionXf" described in the methodology? Is "Normal" the full OpinionXf model?
4/ The paper claims to outperform state-of-the-art models, yet the experiments only compare against internal baselines. A comparison against state-of-the-art LLM would be helpful to substantiate the claims and support the practicality of this problem.

---

### Official Review · Reviewer_ydHu · 2025-11-01

**Soundness:** 1
**Presentation:** 1
**Contribution:** 2
**Rating:** 2
**Confidence:** 4

**Summary:**

This paper presents a comparative study of NLP techniques aimed at modeling deliberative discourse and analyzing opinion shifts. Using a novel dataset composed of diverse viewpoints, the authors simulate deliberation through product presentations designed to elicit changes in audience opinions. They evaluate Frequency-Based Discourse Modulation and the Quantum-Deliberation Framework, which are two of the best performing approaches.

There are several substantial issues that need to be addressed before this manuscript is ready for publication. These include the lack of enough information about the data and experiments, questionable experimental setup, and insufficient literature survey. Please refer to the list of weaknesses for more details.

**Strengths:**

1.	This paper tackles an important issue of opinion shifts post-deliberation.
2.	The authors construct a novel dataset of 100+ survey results pre and post-deliberation, though not enough information has been provided.

**Weaknesses:**

1.	Important details about the dataset are missing: What is the exact number of data points from student surveys? What is the exact number of data points that were generated? What are the questions and answer choices on the survey? How did the professors ensure “plausible reasoning, deliberative patterns, and cognitions consistent with cognitive theories of opinion formation? Did they quantify the evaluation? How many professors were involved in verifying the quality of the data, and what was the rate of agreement? How were LLMs used to generate the synthetic data exactly? Were they assessed for quality in the same manner as the student surveys?
2.  A concern regarding the experimental setup: Is Table 1 reporting the performance on the validation set or the test set? If the former is the case, does that mean the design choices of the architecture and the hyperparameters were optimized for---and also tested on---the same data points? This would inflate the scores, making them unreliable predictors of the model’s performance in the wild.
2.	The survey of literature needs to be expanded quite a bit. Citing 8 papers is not sufficient for adequately situating this work in the context of existing literature. For instance, the paper the new dataset without citing them. Which existing works study opinion shifts with or without deliberation?
3.	The writing needs to be further edited. There is repeated content (e.g. The information about the dataset in Sec 3, 4.1 and 5.) and inconsistencies across sections (e.g. the description of baselines and metrics described in Sec 4.3 does not match those in Sec 5.3). Also, please make references to tables and figures in the main text.

**Questions:**

My suggestion is to address the concerns and questions raised in the weaknesses section.

---

### Official Review · Reviewer_5ZDC · 2025-11-01

**Soundness:** 3
**Presentation:** 3
**Contribution:** 2
**Rating:** 2
**Confidence:** 5

**Summary:**

This paper introduces a computational framework for modeling opinion shifts in deliberative discourse through a combination of frequency based fusion and a quantum inspired deep learning approach. The authors develop a dataset of pre and post exposure survey responses that capture how opinions evolve after structured deliberation. The study compares a baseline transformer, a frequency based fusion variant and a quantum based model with the latter achieving the strongest performance. The work connects deliberation studies and computational modeling by treating opinion change as a dynamic process with implications for persuasion, decision making and societal discourse in NLP contexts.

**Strengths:**

- The paper integrates deliberation theory from social sciences and targets an underexplored region of computational social reasoning.

- The paired dataset is a valuable resource for studying opinion change as it goes beyond traditional stance detection.

- The choice of a frequency domain fusion with a quantum inspired token mechanism are bold and the presentation of comparisons between each model is clear evident by the emphasis in trade off between performance and interpretability.

**Weaknesses:**

- The quantum based model lacks ablation analysis distinguishing the effects of quantum based tokens versus frequency fusion based tokens. It is unclear whether performance gains are from the quantum design or the model signs. An ablation analysis isolating component contributions would help the paper significantly.

- The paper generalizes findings from a limited academic dataset to large scale social applications. The discussion section cites that applications in policy making and social media without validation. It appears as if the claims have been overstated, rephrasing the same as potential applications prevents overstating actual applicability.

- There are no examples from the dataset provided only descriptive summaries of its structure and composition. The confirms restricted access which limits the ability to assess the quality and realism. Releasing an anonymized or synthetic sample aligned with reproducibility standards would strengthen the transparency.

The paper is an ambitious and creative attempt to merge deliberative theory with deep learning. The papers interdisciplinary approach and methods are commendable. The authors should clarify the role of model components through ablation and provide a representative synthetic sample of the dataset. With these improvements the work advances deliberation aware NLP and provides a solid base for future research.

**Questions:**

Please see the concerns raised in the weaknesses section

---

### Meta-Review · Area_Chair_WmLz · 2025-12-26

**Summary:**

No rebuttal was provided. Recommend rejection.

**Reviewer Scores:**

Initial score 2, 2, 2, 2. No rebuttal.

---

### Decision · Program_Chairs · 2026-01-26

Reject